# Ground-state electron transfer in all-polymer donor:acceptor blends enables aqueous processing of water-insoluble conjugated polymers

Tiefeng Liu[1,2], Johanna Heimonen[1,3], Qilun Zhang [1,3], Chi-Yuan Yang [1,4], Jun-Da Huang[1,3,4], Han-Yan Wu[1], Marc-Antoine Stoeckel [1,2,4], Tom P. A. van der Pol [1], Yuxuan Li[5], Sang Young Jeong[6], Adam Marks[7], Xin-Yi Wang[8], Yuttapoom Puttisong [5], Asaminew Y. Shimolo[1,3], Xianjie Liu[1], Silan Zhang [1,3], Qifan Li[1], Matteo Massetti[1], Weimin M. Chen [5], Han Young Woo [6], Jian Pei [8], Iain McCulloch [7], Feng Gao [5], Mats Fahlman [1,3], Renee Kroon[1,3] & Simone Fabiano [1,2,3,4] ✉

Water-based conductive inks are vital for the sustainable manufacturing and widespread adoption of organic electronic devices. Traditional methods to produce waterborne conductive polymers involve modifying their backbone with hydrophilic side chains or using surfactants to form and stabilize aqueous nanoparticle dispersions. However, these chemical approaches are not always feasible and can lead to poor material/device performance. Here, we demonstrate that ground-state electron transfer (GSET) between donor and acceptor polymers allows the processing of water-insoluble polymers from water. This approach enables macromolecular charge-transfer salts with 10,000× higher electrical conductivities than pristine polymers, low work function, and excellent thermal/solvent stability. These waterborne conductive films have technological implications for realizing high-performance organic solar cells, with efficiency and stability superior to conventional metal oxide electron transport layers, and organic electrochemical neurons with biorealistic firing frequency. Our findings demonstrate that GSET offers a promising avenue to develop water-based conductive inks for various applications in organic electronics.

Conjugated polymers combine the easy solution processability, mechanical flexibility, lightweight, and versatile chemical synthesis of polymers with the electrical proprieties of traditional semiconductors[1–4]. These unique characteristics make them highly valuable for applications in various industries, such as displays[5–7], energy harvesting/storage[8–11], sensing[12–14], and healthcare[15–17]. However, the conventional processing methods of conjugated polymers often involve hazardous solvents like chlorinated, aromatic, or highly volatile solvents, which can be toxic, flammable, and harmful to people and the environment. This poses a significant challenge to the broad commercial and sustainable implementation of conjugated polymers and relative electronic devices. Water still represents the safest and most sustainable option for large-scale printing of electronics. As a result, developing waterborne processing methods for conjugated polymers has become crucial.

A common strategy to disperse conjugated polymers in aqueous media is to confine them in the form of micro-/nanoparticles[18–21]. Due to the intrinsic hydrophobic nature of most conjugated polymers, surfactants and other additives are needed to

reach the appropriate particle uniformity and stabilize the aqueous dispersion[22–27]. In addition, while conjugated polymers can be dispersed in suitable solvents at low concentrations, higher polymer loading in the g l$^{-1}$ range–needed for the subsequent solution processing step–often leads to particle aggregation. Thus, highly concentrated aqueous dispersions of conjugated polymers necessitate the use of additives to counteract aggregation[28,29] or polyelectrolytes to induce the formation of complex coacervates[30,31]. Polyelectrolytes have also been employed in the synthesis of conjugated polymers in water to stabilize their resulting aqueous dispersions. This includes the prototypical poly(3,4-ethylenedioxythiophene):polystyrene sulfonate (PEDOT:PSS)[32,33]. In this instance, the insulating PSS chain acts as a template for the chemical polymerization of EDOT monomers in the presence of oxidants while providing immobile counterions to compensate for the holes on the PEDOT chains[34,35]. However, typical surfactants or polyelectrolytes are insulators that remain in the solid films, adversely impacting materials' electrical properties and relative devices' metrics[36,37]. Therefore, post-processing treatments are necessary to enhance their electrical performance[38]. To circumvent the use of surfactants and polyelectrolytes, the conjugated polymer backbone can be chemically modified by introducing hydrophilic side chains that enhance solubility in polar solvents[39,40], thereby facilitating the formation of aqueous dispersions. While this chemical approach is effective in enhancing the overall hydrophilicity of the system, it is not always feasible, as introducing hydrophilic side chains is either challenging or results in poor material properties[41,42].

Here, we demonstrate that mutual electrical doping in donor: acceptor polymer blends aids in the dispersion of water-insoluble conjugated polymers in aqueous solutions. We show that when a water-soluble donor polymer with low ionization energy (IE) is mixed with a high-electron-affinity (EA) water-insoluble acceptor polymer, it can undergo a spontaneous ground-state electron transfer (GSET[43], Fig. 1a) which promotes dispersion of the latter. This mutual electrical doping also results in a 10,000-fold increase in electrical conductivity compared to pristine polymers. This approach yields a macromolecular charge-transfer salt that is distinct from PEDOT:PSS, as it

does not rely on oxidative agents to trigger the polymerization of the conducting polymeric phase, and both donor and acceptor polymers could assist in the transport of charge carriers. In addition, we show that this strategy is general and can be applied to other high-performance water-insoluble conjugated polymers. These conductive, water-processable inks are applied as the electron transport layers in organic solar cells, with conversion efficiency and operational stability surpassing traditional zinc oxide layers. We also used them to develop high-performance organic electrochemical transistors, ultra-low power complementary electrochemical inverters, and organic electrochemical neurons with biorealistic spiking frequencies. We anticipate that the use of GSET to produce water-based conductive inks will provide a practical solution when traditional chemical modification methods are not viable, contributing to the widespread commercial and sustainable implementation of organic electronics.

## Results

### Ground state electron transfer

Poly(benzimidazobenzophenanthroline) (BBL) was chosen as an example of a water-insoluble conjugated polymer with remarkable electron mobility and a high EA (Fig. 1a)[44–47]. Because of its rigid polymer backbone and lack of solubilizing polar side chains, BBL is not soluble in any common solvents and quickly aggregates in water (Fig. 1b), resulting in coarse films when spin-coated onto a substrate (Fig. 1c). Common surfactants do not aid the solubilization of BBL in water (Supplementary Figs. 3–5). However, when the water-soluble conjugated polyelectrolyte poly(potassium 3-(2-(2-(2-(thiophen-3-yloxy)ethoxy)ethoxy)ethoxy)propanoate)-2,5-thiophene-diyl) (PCAT-K) is added to a water-based BBL particle dispersion, it yields a homogeneous BBL:PCAT-K solution (Fig. 1b), which can be spin-cast to form uniform films (Fig. 1c). The addition also leads to a significant reduction in particle size from ∼530 nm for pure BBL to about 30 nm for BBL:PCAT-K (Fig. 1d). This results in smoother films compared to pure BBL processed from water, as visible by atomic force microscopy analysis (Supplementary Fig. 6). Grazing-incidence wide-angle X-ray scattering (GIWAXS) also shows that the BBL:PCAT-K films have a weaker diffraction pattern compared to pure BBL films, indicative of

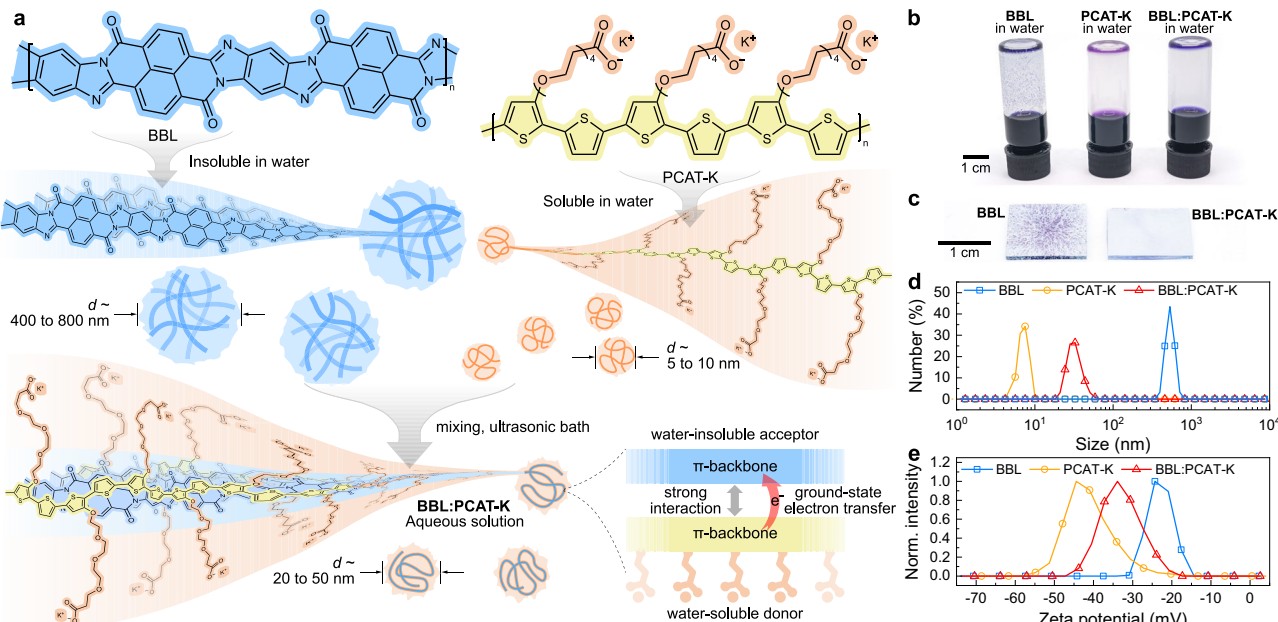

**Fig. 1 | GSET-assisted aqueous processing of BBL. a** Chemical structure of BBL and PCAT-K. BBL is insoluble in water and aggregates, forming particles with dimensions of 400–800 nm. PCAT-K is soluble in water and forms nanoparticles of 5–10 nm. When BBL and PCAT-K are mixed, they undergo a spontaneous GSET, yielding a homogeneous dispersion in water. **b** Photographs of BBL, PCAT-K, and BBL:PCAT-K dispersion in water and relative spin-cast thin films on glass **c**. Particle size **d** and zeta potential **e** of BBL, PCAT-K, and BBL:PCAT-K in solution.

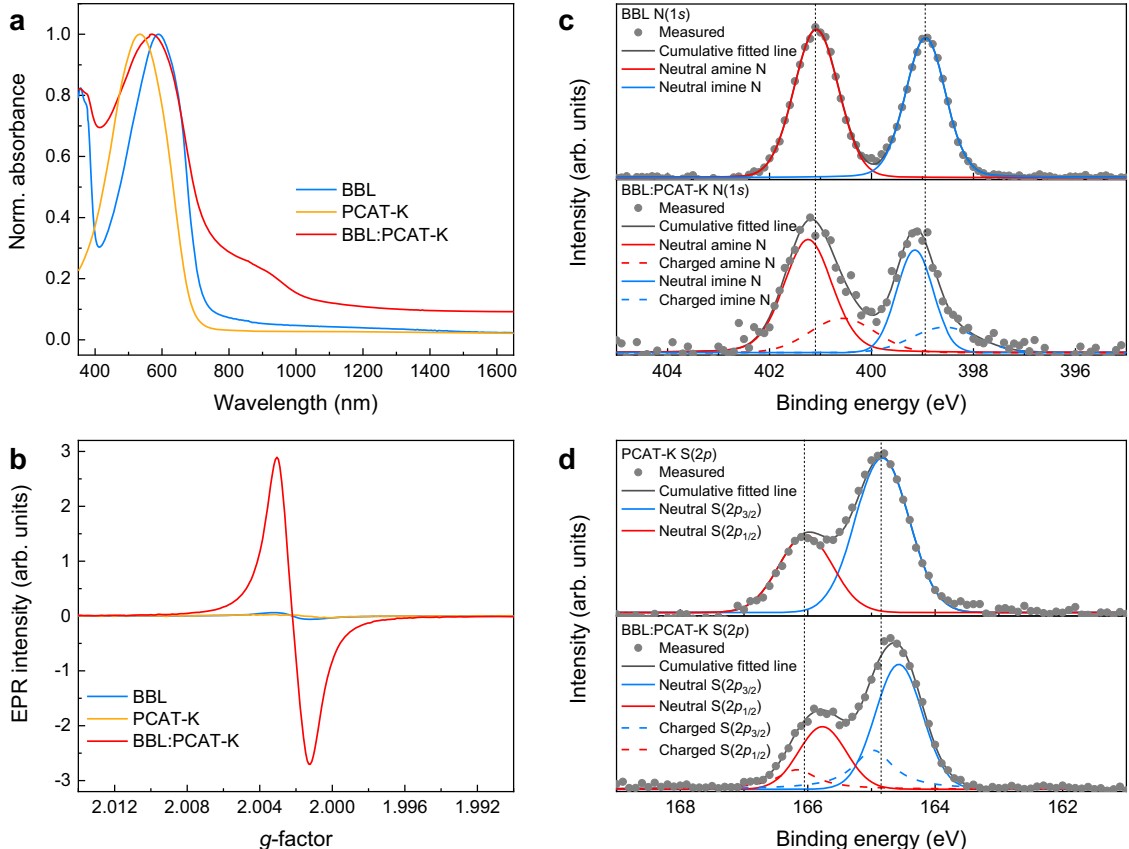

**Fig. 2 | Characterization of the GSET effect in the BBL:PCAT-K blend. a** Normalized absorbance of BBL, PCAT-K, and BBL:PCAT-K thin films. **b** EPR spectra of BBL, PCAT-K, and BBL:PCAT-K thin films. **c**, **d** XPS spectra of BBL, PCAT-K, and BBL:PCAT-K thin films: **c** N(1$s$) and **d** S(2$p$).

good polymer intermixing (see Supplementary Figs. 7 and 8 for further discussion). The BBL:PCAT-K solution is stable for over 6 months under ambient conditions, showing only minor aggregation (Supplementary Fig. 9), which we attributed to electrostatic repulsion between BBL:PCAT-K nanoparticles[48]. This repulsion originates from the dissociation of the carboxylic groups in PCAT-K, forming negative charges on the surface of the nanoparticles, as indicated by an average zeta potential of −34 mV (Fig. 1e).

The UV-vis-NIR absorption spectra of the water-processed BBL:PCAT-K blend films reveal a prominent peak in the range 700–1100 nm, which is ascribed to the presence of polarons in the doped polymers (Fig. 2a). This distinctive feature is only observed in the absorption spectra of the blend films, indicating that polarons are generated when BBL and PCAT-K are mixed. We refer to Supplementary Fig. 10 for a detailed analysis of the UV-vis-NIR absorption spectrum of the BBL:PCAT-K blend and its comparison with individually (chemically) doped BBL and PCAT-K films. The formation of polarons is also confirmed by electron paramagnetic resonance (EPR) spectroscopy, revealing the presence of EPR active species in the all-polymer blend films (Fig. 2b and Supplementary Table 1, see Supplementary Fig. 11 for the EPR spectra of the BBL/PCAT-K bilayer).

X-ray photoelectron spectroscopy (XPS) analysis of the blend films reveals the presence of both n-doped BBL and p-doped PCAT-K (Fig. 2c, d). Pristine BBL films exhibit two distinct peaks at 398.4 eV and 400.5 eV, corresponding to the two different nitrogen sites[49]. Pristine PCAT-K films show two peaks at 164.1 eV and 165.3 eV, related to the spin-split doublet of sulfur in the thiophene ring. After mixing, two new peaks at lower binding energy appear in the nitrogen N(1$s$) spectrum, while two new peaks at higher binding energy appear in the sulfur S(2$p$) spectrum. The shifts towards lower/higher binding energies in the position of these new peaks are indicative of the presence of negatively charged BBL and

positively charged PCAT-K chains. From the peak areas, we estimated a doping level of 28% for both BBL and PCAT-K. Fourier-transformed infrared (FTIR) spectra of the BBL:PCAT-K blend, and their comparison with individually (chemically) doped BBL and PCAT-K films, also provide evidence of doped BBL and PCAT-K chains (Supplementary Fig. 12).

The presence of both positive and negative polarons in the BBL:PCAT-K polymer blend is consistent with a GSET between the low-IE PCAT-K donor polymer and the high-EA BBL acceptor polymer. We then performed ultraviolet photoelectron spectroscopy (UPS) to measure their energy level shift to confirm mutual electrical doping in this donor:acceptor polymer blend. As shown in Supplementary Fig. 13, the work function (WF) of pristine BBL films is about 4.63 eV. After depositing PCAT-K (IE$_{PCAT-K}$ = −4.73 eV), the BBL/PCAT-K bilayer shows a clear shift in the secondary electron cut-off of the UPS spectra. This is in agreement with an interface vacuum level shift due to electron transfer from PCAT-K to BBL (Supplementary Fig. 14)[50]. To further corroborate the observation that GSET is indeed responsible for the dissolution of BBL in water, we investigated another donor polymer with similar chemical structures to PCAT-K, namely poly[3-(potassium-5-pentanoate)thiophene-2,5-diyl] (P3CPT-K, IE$_{P3CPT-K}$ = −5.03 eV). Because of the higher IE, P3CPT-K yields no significant GSET when in contact with BBL, as demonstrated by the near-negligible vacuum level shift at the interface (Supplementary Figs. 13 and 14). The optical images of BBL:P3CPT-K dispersion and the corresponding UV-vis-NIR absorbance are presented in Supplementary Fig. 15, corroborating the absence of GSET and resulting poor water solubility.

## Electrical properties of BBL:PCAT-K blends

The electrical conductivity of BBL:PCAT-K thin films varies with the BBL content in the blend, as shown in Fig. 3a. The conductivity increases from less than $10^{-5}$ S cm$^{-1}$ for the pristine BBL and PCAT-K films to over

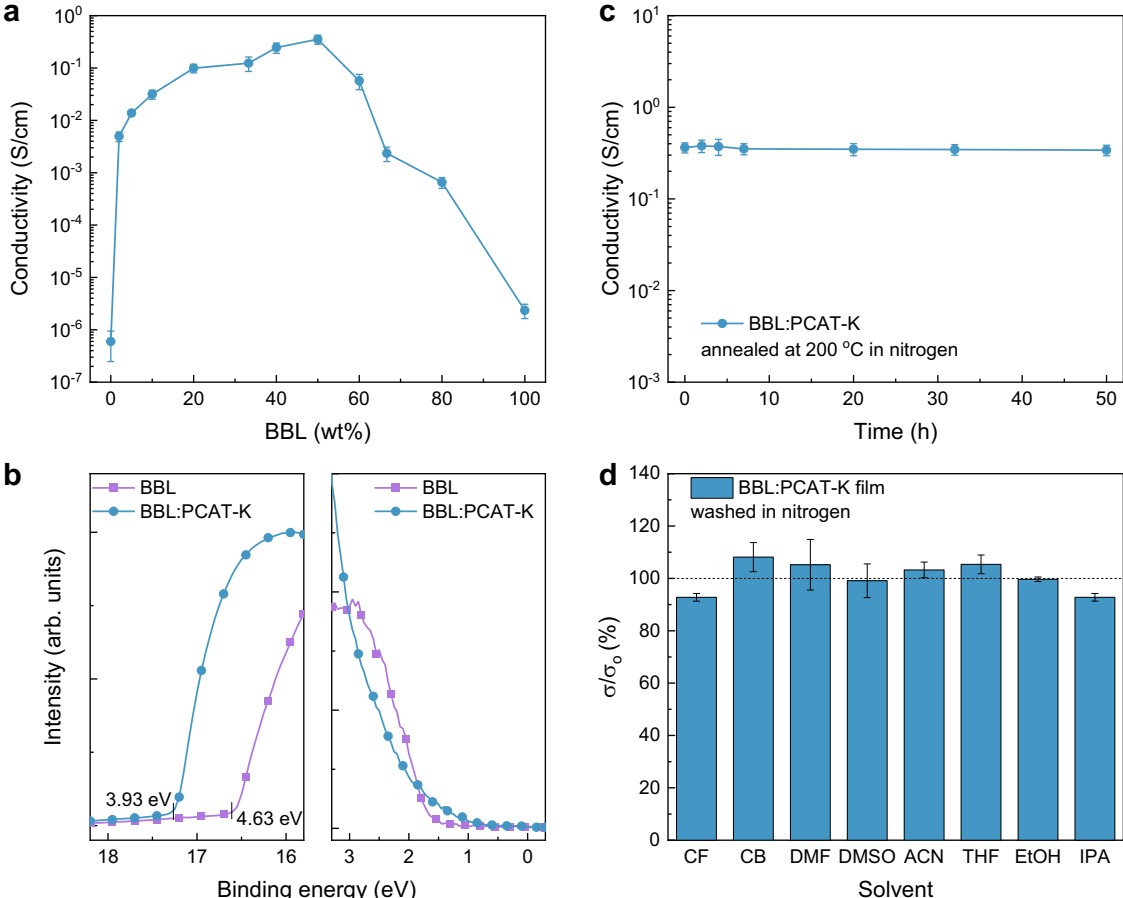

**Fig. 3 | Electrical properties and stability of the BBL:PCAT-K blends.**
**a** Conductivity of BBL:PCAT-K films as a function of BBL content. **b** UPS spectra of BBL and BBL:PCAT-K (1:1) films. **c** Thermal stability of BBL:PCAT-K (1:1) films annealed at 200 °C in nitrogen for 50 h. **d** Stability of BBL:PCAT-K (1:1) films washed with common organic solvents (CF chloroform, CB chlorobenzene, DMF dimethylformamide, DMSO dimethylsulfoxide, ACN acetonitrile, THF tetrahydrofuran, EtOH ethanol, and IPA isopropyl alcohol). Error bars indicate the SD of ten experimental replicates.

$10^{-1}$ S cm$^{-1}$ for BBL:PCAT-K films, with a maximum conductivity of $0.35 \pm 0.06$ S cm$^{-1}$ achieved at 50 wt% BBL content. The blend films retain their maximum conductivity when stored in an inert atmosphere for over one month (Supplementary Fig. 16). Additionally, the conductivity is independent of film thickness <350 nm but slightly decreases to about 0.12 S cm$^{-1}$ for μm-thick films (Supplementary Fig. 17). In contrast, the conductivity of BBL:P3CPT-K blend films and potassium acetate doped BBL is about $10^{-4}$ S cm$^{-1}$ (Supplementary Fig. 18), which is more than 100× lower than BBL:PCAT-K. The BBL:PCAT-K blends exhibit a negative Seebeck coefficient, regardless of the BBL content, suggesting that the blend conductivity is dominated by the electron transport in the n-type polymer BBL (Supplementary Fig. 19). Additionally, the BBL:PCAT-K films show a low WF of 3.93 eV (Fig. 3b), which is beneficial for electron extraction/injection in organic optoelectronic devices. It is important to note that the electrical conductivity of the blend films can be further improved to $2.3 \pm 0.08$ S cm$^{-1}$ by adding 33 wt% poly(ethyleneimine) (PEI) into the aqueous solution (Supplementary Fig. 20). The amine-based PEI acts as a polymeric dopant and donates electrons to the BBL polymer backbone[21]. These conductivity values are remarkable for n-type polymeric inks processed from water and on par with those of commercial water-based p-type polymeric inks like PEDOT:PSS PH1000 formulations, which show conductivities of about 1 S cm$^{-1}$ without additives.

The BBL:PCAT-K films also show remarkable thermal stability, with no significant degradation of the conductivity even after 50 h of continuous thermal annealing at 200 °C in an inert atmosphere (Fig. 3c). This suggests that the interaction between BBL and PCAT-

K is strong and can withstand high temperatures. Additionally, the films are compatible with various organic solvents (Fig. 3d), typically used in organic electronics[51,52]. This compatibility means the films can be used in multi-layered film fabrication for organic electronic devices such as solar cells and light-emitting diodes without affecting their conductivity.

## Generality of the GSET-induced solubilization strategy
The effectiveness of the GSET-promoted solubilization of conjugated polymers in water is demonstrated for other water-insoluble semiconductors, including several polymers such as the side chain-free ladder-type polymers poly(4-aza-benzimidazobenzophenanthroline) (PyBBL) and tetrachlorinated poly(benzimidazoanthradiisoquinolinedione) (Cl$_4$-BAL), or lactam-based and benzodifurandione-based polymers bearing side chains with different hydrophilicity. When mixed with PCAT-K, both side chain-free ladder-type polymers PyBBL[53] and Cl$_4$-BAL[54] disperse in water, yielding homogeneous solutions (Supplementary Fig. 21).

Next, we varied the side-chain hydrophilicity/hydrophobicity of the water-insoluble high-EA conjugated polymers to understand their impact on the GSET-promoted solubilization process. To do this, we selected the fluorinated benzodifurandione-based poly(p-phenylene vinylene) (FBDPPV) functionalized with oligo ethylene glycol side chains (FBDPPV-OEG) or long 4-octadecyldocosyl side chains (FBDPPV-4OD)[55]. Remarkably, we noticed that FBDPPV-OEG dissolved in water at a high concentration (20 mg ml$^{-1}$) when mixed with PCAT-K

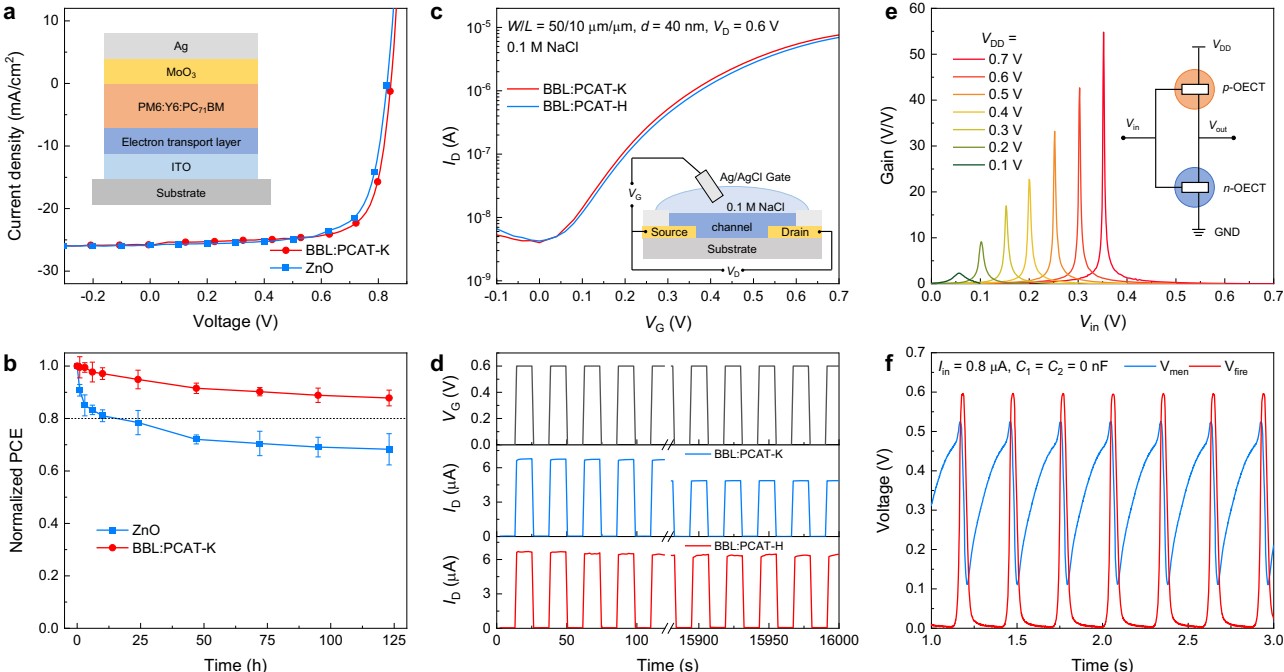

**Fig. 4 | Application of BBL:PCAT-K in organic electronic devices. a** $J$–$V$ curves of OSCs with the structure ITO/ETL/PM6:Y6:PC$_{71}$BM/MoO$_3$/Ag, where the ETL is either BBL:PCAT-K or ZnO. **b** Operational stability of OSCs with BBL:PCAT-K and ZnO as ETL under continuous light irradiation at AM 1.5 G in an N$_2$-filled glovebox. Error bars indicate the SD of eight experimental replicates. **c** Transfer curves and **d** long-term stability tests of BBL:PCAT-K and crosslinked BBL:PCAT-H based OECT. **e** Voltage gains of the inverter at various supply voltages ($V_{DD}$ from 0.1 to 0.7 V). **f** Spiking behaviors of the organic electrochemical neurons with $C_{mem} = C_f = 0$ nF, input current of 0.8 µA, and $V_{DD}$ of 0.6 V. The OSC, OECT, and complementary inverter structures are reported as insets in **a**–**d**, respectively.

(Supplementary Fig. 22). This is in stark contrast to what is observed for FBDPPV-4OD, which traps PCAT-K out of the aqueous solution, likely due to the high hydrophobicity of its long alkyl side chains (Supplementary Fig. 23). We also tested a series of lactam polymers p(g$_7$NC$_n$N) functionalized with a mixture of oligo ethylene glycol side chains and alkyl chains with gradually increasing length[56]. Interestingly, we observed that polymers with shorter alkyl chains (i.e., p(g$_7$NC$_2$N) and p(g$_7$NC$_4$N)) dissolve well in the PCAT-K aqueous solution, while longer alkyl chains (i.e., p(g$_7$NC$_6$N) and p(g$_7$NC$_8$N)) cause the polymers to crash out of the PCAT-K aqueous solution (Supplementary Fig. 24). We attributed this effect to the increased hydrophobicity of the polymer bearing longer alkyl chains, inferred from contact angle measurements (Supplementary Fig. 25).

**Applications of the water-based BBL:PCAT-K blend**

Next, we demonstrate the potential of these water-based conductive inks as active layers in prototypical electronic devices. First, we fabricated organic solar cells (OSCs) with the BBL:PCAT-K films used as the electron transport layer in an inverted OSC structure (see Fig. 4a and details about the materials and device fabrication steps in the Methods section). Typical current density–voltage ($J$–$V$) curves of an ITO/BBL:PCAT-K/PM6:Y6:PC$_{71}$BM/MoO$_3$/Ag OSC under simulated AM 1.5 illumination (100 mW cm$^{-2}$) are reported in Fig. 4a. The optimized devices show an open-circuit voltage $V_{OC} = 0.84$ V, fill factor FF = 0.74, short-circuit current density $J_{SC} = 25.74$ mA cm$^{-2}$, and power conversion efficiency PCE = 16.03%. These values are comparable to those measured for the reference devices comprising ZnO as the electron transport layer: $V_{OC} = 0.83$ V, FF = 0.72, $J_{SC} = 25.85$ mA cm$^{-2}$, and PCE = 15.44%. The current density was confirmed by external quantum efficiency (EQE) measurements (see Supplementary Fig. 26). Similar results are obtained using other polymeric donors and non-fullerene acceptors (Supplementary Fig. 27), which are summarized in Supplementary Table 2.

Remarkably, BBL:PCAT-K-based devices retain almost 90% of their initial performance after 123 h of continuous illumination at 100 mW cm$^{-2}$ in nitrogen (Fig. 4b). For comparison, the reference ZnO-based devices degrade quickly, with more than 20% loss of their initial performance after only 24 h of illumination, due to the well-known instability of the ZnO layer[57]. The photovoltaic parameters under continuous illumination and $J$–$V$ characteristics on different light intensities are reported in Supplementary Figs. 28-29. Supplementary Table 3 provides a summary of the water-processable conjugated polymers that have been reported as electron transport layers in OSCs.

We also tested BBL:PCAT-K as an n-type organic mixed ionic-electronic conductor for application in organic electrochemical transistors (OECTs, Fig. 4c). When in contact with the aqueous electrolyte, charges in BBL:PCAT-K are depleted or trapped, so the films become non-conductive (Supplementary Fig. 30) and the resulting OECTs operate in accumulation mode. Fig. 4c shows the typical transfer characteristics of a BBL:PCAT-K (1:1)-based OECT. The source-drain current is minimal at zero gate voltage ($V_G$) and increases by over three orders of magnitude when a positive $V_G$ is applied. We observed no current when a negative $V_G$ was applied, confirming the blend's unipolar n-type character. These BBL:PCAT-K-based OECTs exhibit electrical performance comparable to conventional BBL-based OECTs processed from methanesulfonic acid (see Supplementary Fig. 31 and Supplementary Table 4 for a survey of the OECT characteristics of different BBL formulations reported in the literature). As PCAT-K is soluble in water, BBL:PCAT-K-based OECTs degrade upon continuous testing (Fig. 4d). This can be overcome by physically crosslinking PCAT-K through hydrogen-bonding, as the pKa of the carboxylate (~5) allows for protonation with dilute acid solutions[58]. The acid treatment converts the PCAT-K to its insoluble form, which we designated PCAT-H (Supplementary Fig. 32). The crosslinked BBL:PCAT-H-based OECTs show similar

device performance to the non-crosslinked BBL:PCAT-K (Fig. 4c and Supplementary Figs. 33 and 34), but most importantly, they show no current degradation after nearly 5 h of continuous operation (Fig. 4d). The OECTs' output characteristics are reported in Supplementary Fig. 35.

We then fabricated OECT-based complementary inverters with BBL:PCAT-K as the n-type OECT and the polythiophene P(g$_4$2T-T)[59] as the p-type OECT (see Fig. 4e and details about the inverter fabrication in the Methods section). By optimizing the p-/n-type OECTs' characteristics (Supplementary Fig. 36), we achieved complementary inverters operating at supply voltages ($V_{DD}$) as low as 0.1 V, with a switching threshold ($V_M$) of 0.055 V, voltage gains of 2.3 V V$^{-1}$, static power consumption ($P_{static}$) of 0.32 nW, and dynamic power consumption ($P_{dynamic}$) of 0.41 nW. A maximum gain of 54.8 V V$^{-1}$ was achieved at $V_{DD} = 0.7$ V with a $P_{dynamic} < 1.94$ μW (see Fig. 4e, Supplementary Fig. 37, and Supplementary Table 5). With these complementary inverters at hand, we fabricated leaky integrate-and-fire type spiking organic electrochemical neurons[60] (OECNs, Supplementary Fig. 38), reaching biorealistic firing frequencies of about 10 Hz (Supplementary Figs. 39 and 40).

## Discussion

In summary, we developed a strategy to simultaneously enhance the solubility and conductivity of conjugated polymers in water. We showed that GSET in all-polymer donor:acceptor blends promotes the dispersion of water-insoluble conjugated polymers in aqueous solutions. This strategy is general and can be applied to other water-insoluble acceptor polymers. This mutual electrical doping yields solid films with 10,000× higher electrical conductivities than pristine polymers. It also results in thin films having a low work function and excellent thermal and solvent stability. We utilized these waterborne formulations to fabricate electron transport layers in OSCs incorporating non-fullerene acceptors, resulting in power conversion efficiencies and operational stability that outperformed those obtained with ZnO as the traditional electron transport layer. We also demonstrated high-performance OECTs, ultra-low power complementary OECT-based inverters, and OECNs with biorealistic spiking frequencies. Using GSET to produce water-based conductive inks is expected to offer a practical solution in cases where conventional chemical modification methods are not feasible. This advancement is poised to significantly contribute to the broad adoption and sustainable implementation of organic electronics in commercial applications.

## Methods

### Materials

BBL ($M_v = 23$ kDa) was synthesized by a polycondensation reaction of 1,4,5,8-naphthalenetetracarboxylic dianhydride and 1,2,4,5-tetra-aminobenzene tetrahydrochloride in poly(phosphoric acid) at high temperature[61]. PCAT-K was synthesized using the route reported in Supplementary Information ($M_n = 12$ kDa and polydispersity of 3.8). PyBBL, Cl$_4$-BAL, FBDPPV-OEG, FBDPPV-4OD, and P(g$_7$NC$_n$N) were synthesized as reported previously[53,54,56,62,63]. P3CPT-K was prepared by dissolving P3CPT (Rieke Metals Inc, $M_w = 55$ kDa) in a KOH aqueous solution with an equivalent ratio. Surfactants were purchased from Sigma-Aldrich, while the organic active materials in OSCs were purchased from 1-Material Inc. All the solvents were purchased from Sigma-Aldrich. The viscosity-average molecular weight ($M_v$) of BBL was estimated by measuring the intrinsic viscosity ($\eta$) of BBL in methanesulfonic acid (MSA). The $\eta$ and $M_v$ follow the Mark-Houwink-Sakurada equation $\eta = KM_v^{\alpha}$, where $K = 5.11 \times 10^{-6}$ g dl$^{-1}$ and $\alpha = 1.34$. The viscosity was measured with an Ubbelohde-type viscometer. The $M_n$ of PCAT-K was measured by gel permeation chromatography (Agilent 1280 Infinity system).

### Sample preparation

Water-based BBL dispersion was prepared by a solvent-exchange method. In brief, the BBL was first dissolved in MSA to form a deep red solution (5 mg ml$^{-1}$). Then, the BBL-MSA solution was added dropwise to IPA to remove MSA under stirring (1000 r.p.m.), which generated dark purple BBL nanoparticles suspension. The BBL nanoparticles were washed with water until pH = 7. The homogeneous water-based BBL:PCAT-K ink was prepared by mixing BBL nanoparticles and PCAT-K and sonicated (180 W) for over 30 min. Sonication does not affect the polymers' molecular weight (see Supplementary Fig. 41). The highest concentration of the ink is about 8 mg ml$^{-1}$ (BBL:PCAT-K 1:1). The weight ratio of BBL in the homogeneous solution is less than 60%. We observed that high molecular weight PCAT-K can help disperse more BBL, while high molecular weight BBL requires more PCAT-K to disperse effectively in water. The relationship between the BBL:PCAT-K ratio and molecular weight of the polymers is reported in Supplementary Fig. 42. The BBL:PCAT-K film can be deposited by spin-coating and first annealed at 120 °C in air for 5 min and then annealed at 150 °C in an N$_2$-filled glovebox for 1 h to remove any traces of water for high conductivity. The film thickness was controlled by adjusting the concentration: 2 mg ml$^{-1}$ for 10 nm and 6 mg ml$^{-1}$ for 50 nm.

### UPS and XPS spectroscopy

UPS and XPS experiments were carried out using a Scienta ESCA 200 system equipped with an SES 200 electron analyzer, a monochromatic Al Ka X-ray source (1486.6 eV), and a helium discharge lamp (21.22 eV) for XPS and UPS in a vacuum of $1 \times 10^{-10}$ mbar, respectively. All spectra were collected at normal emission and were calibrated by a sputter-cleaned Au film with the Fermi level at 0 eV and the Au(4$f$) peak at 84.0 eV. The work function of the films was extracted from the edge of the secondary electron cutoff in UPS by applying a −3 V bias on the sample.

### Electron paramagnetic resonance spectroscopy

X-band EPR was performed using a Bruker ELEXSYS E500 spectrometer operating at about 10 GHz. The EPR signal was recorded in the dark at room temperature. Quantitative EPR was calibrated by a standard sample.

### Thin-film morphology characterization

The morphology was measured by AFM (Bruker Dimension Icon XR) in tapping mode using a silicon nitride cantilever having a spring constant of 40 N m$^{-1}$. GIWAXS experiments were performed at Beamline 9 A at the Pohang Accelerator Laboratory in South Korea. The X-ray energy was 11.07 eV, and the incidence angle was 0.12°. Samples were measured in a vacuum, and the total exposure time was 10 s. The scattered X-rays were recorded by a charge-coupled device detector located 221.7788 mm from the sample.

### Dynamic Light Scattering

The solutions' particle size and zeta potential were characterized by dynamic light scattering with a Zetasizer Nano ZS90 (laser wavelength $\lambda = 632.8$ nm) at room temperature. All solutions are diluted to about 0.1 mg ml$^{-1}$ (without filtering) and sonicated for 5 min before testing. The results are calculated by averaging measurements from ten replicates.

### UV-vis-NIR and FTIR absorption spectroscopy

The UV-vis-NIR absorption spectra were collected using a PerkinElmer Lambda 900 spectrophotometer with a resolution of 2 nm. The FTIR spectra were collected in transmission with a Bruker Equinox 55 spectrometer averaging 100 scans with a resolution of 4 cm$^{-1}$ and a zero-filling factor of 2. All measurements were performed inside an air-tight sample holder, sealed in an N$_2$-filled glovebox. The films were prepared on calcium fluoride windows by spin-coating.

## Electrical characterization

Electrical conductivity and Seebeck coefficient measurements were performed inside an $N_2$-filled glovebox using a Keithley 4200-SCS semiconductor characterization system. The channel length/width is 30 μm/1000 μm for electrical characterization while 0.5 mm/15 mm for Seebeck coefficient characterizations. The chromium/gold (5/50 nm) electrodes were thermally evaporated on cleaned glass substrates through a metal shadow mask.

## OSC fabrication and characterization

The OSCs were fabricated in the inverted ITO/ETL/active/$MoO_3$/Ag structure. Before depositing ETL films, the pre-cleaned ITO electrode was treated with $O_2$ plasma (100 W) for 5 min. The BBL:PCAT-K ETL was deposited in ITO electrodes with the same procedure mentioned above. The thickness of BBL:PCAT-K ETL is about 10 nm. The ZnO ETL was prepared via a traditional sol-gel method[64]. In brief, 1 g zinc acetate dihydrate and 0.28 g ethanolamine were added to 10 ml methoxyethanol and stirred for over 24 h to yield the ZnO precursor. The ZnO film was deposited on the ITO electrode by spin-coating from precursor at 3000 r.p.m. and annealed at 200 °C for 30 min in the air. Then, about 100 nm thickness organic active layer was spin-coated on the ITO/ETL from solution and then annealed at 100 °C for 10 min in an $N_2$-filled glovebox. The PM6:Y6:$PC_{71}$BM solution was prepared by dissolving in CF (with 0.5 vol% of 1-chloronaphthalene as additive) with the weight ratio of 7:7:1.4 mg ml$^{-1}$. The PM6:BTP-eC9:$PC_{71}$BM solution was prepared by dissolving in CB (with 0.5 vol% of 1,8-diiodooctane as additive) with the weight ratio of 8.5:8.5:1.7 mg ml$^{-1}$. The D18-Cl:N3:$PC_{71}$BM solution was prepared by dissolving in CF (with 0.3 vol% of 1-chloronaphthalene as additive) with the weight ratio of 5:7:1 mg ml$^{-1}$. Finally, the 10 nm/100 nm $MoO_3$/Ag electrode was thermally evaporated with a shadow mask under a vacuum of $1 \times 10^{-6}$ Torr. The effective area of the device is about 4.43 mm$^2$. The device's $J$–$V$ characteristics were measured with a Paios platform under AM 1.5 G illumination (100 mW cm$^{-2}$) from a solar simulator in the $N_2$-filled glovebox. The devices were measured with a step voltage of 0.02 V. The light intensity was determined by a standard silicon photodiode. As for the light intensity-dependent measurements, the light source was a white light-emitting diode integrated with the Paios platform. The EQE spectrum was collected with a Newport Merlin lock-in amplifier.

## OECT, inverter, and OECN fabrication and characterization

OECTs were fabricated following the procedure reported previously[61]. In brief, gold source and drain electrodes were deposited on glass substrates and coated by a 4 μm-thick Parylene C layer. The active material in the channel was deposited by spin-coating and patterned using another layer of Parylene C to define the OECT channel geometry. Finally, a 0.1 M NaCl aqueous solution was used as the electrolyte, and an Ag/AgCl pellet was dipped into the electrolyte to serve as the gate electrode. The crosslinked BBL:PCAT-H was obtained by dipping BBL:PCAT-K film in citric acid solution (10 mg ml$^{-1}$ in IPA) for 1 min and then washing it with IPA to remove the residual acid. The complementary inverters and OECNs were fabricated by connecting a P($g_{42}$T-T) based p-type OECT and a BBL:PCAT-K based n-type OECT using silver paint. The OECTs, inverters, and OECNs were characterized by Keithley 4200A-SCS.

## Reporting summary

Further information on research design is available in the Nature Portfolio Reporting Summary linked to this article.

## Data availability

The authors declare that the main data supporting the findings of this study are available within the paper and its Supplementary Information files. The main data generated in this study are provided in the Supplementary Information/Source Data file. Source data are provided with this paper.

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

## Acknowledgements

This work was financially supported by the Knut and Alice Wallenberg Foundation (2021.0058, 2021.0230, 2022.0034, WWSC, and Wallenberg Initiative Materials Science for Sustainability WISE), the Swedish Research Council (2020-03243, 2020-04538, 2022-04553), Olle Engkvists Stiftelse (204-0256), VINNOVA (2020-05223), the European Commission through the MITICS (GA-964677), HORATES (GA-955837), and SUNREY (GA-101084422) projects, and the Swedish Government Strategic Research Area in Materials Science on Functional Materials at Linköping University (Faculty Grant SFO-Mat-LiU 2009-00971). Y.P. acknowledges the support from the Swedish Energy Agency (EM48594-1). H.Y.Woo acknowledges the financial support by the National Research Foundation (NRF) of Korea (2019R1A6A1A11044070).

## Author contributions

T.L. developed and characterized the polymer blends, performed the electrical measurements, and measured the UV-vis-NIR spectra. J.H. and R.K. designed and synthesized PCAT-K, performed structural analysis and developed its crosslinking chemistry. A.Y.S. assisted with the synthesis of PCAT-K with different molecular weights. J.-D.H. and Q.L. synthesized BBL, PyBBL, and $Cl_4$-BAL. X.-Y.W. and J.P. synthesized FBDPPV-4OD and PBDPPV-OEG. A.M. and I.M. synthesized $P(g_7NC_nN)$. Q.Z., X.L., and M.F. recorded and analyzed the UPS and XPS spectra. M.A.S. recorded and analyzed the AFM. S.Z. and T.P.A.v.d.P. recorded and analyzed the FTIR data. T.P.A.v.d.P. assisted with the analysis of the UV-vis-NIR spectra. S.Y.J., C.-Y.Y., and H.Y.Woo measured and analyzed the GIWAXS data. Y.P. and W.M.C. performed and analyzed the EPR data. T.L., Q.Z., Y.L., and F.G. fabricated and characterized the OSCs. T.L., H.-Y.Wu, and M.M. fabricated and characterized the OECTs, inverters, and neurons. S.F. conceived and designed the project. T.L. and S.F. wrote the manuscript. All authors contributed to the discussion and manuscript preparation.

## Funding

## Competing interests

T.L., J.H., C.-Y.Y., R.K., and S.F. filed provisional patent applications related to this work (application nos. PCT/EP2023/053382, PCT/EP2023/ 053389, PCT/EP2023/053390, filed 10/02/2023). C.-Y.Y. and S.F. are the co-founders of n-Ink AB. The other authors declare that they have no competing interests.

## Additional information

[1]Laboratory of Organic Electronics, Department of Science and Technology, Linköping University, Norrköping, Sweden. [2]Wallenberg Initiative Materials Science for Sustainability, Department of Science and Technology, Linköping University, Norrköping, Sweden. [3]Wallenberg Wood Science Center, Linköping University, Norrköping, Sweden. [4]n-Ink AB, Norrköping, Sweden. [5]Electronic and Photonic Materials, Department of Physics, Chemistry, and Biology, Linköping University, Linköping, Sweden. [6]Department of Chemistry, College of Science, Korea University, Seoul, Republic of Korea. [7]Department of Chemistry, University of Oxford, Oxford, UK. [8]Beijing National Laboratory for Molecular Sciences (BNLMS), Key Laboratory of Polymer Chemistry and Physics of Ministry of Education, Center of Soft Matter Science and Engineering, College of Chemistry and Molecular Engineering, Peking University, Beijing, China. ✉e-mail: simone.fabiano@liu.se

