## [Peer Review File · Nature Communications]

REVIEWER COMMENTS

Reviewer #1 (Remarks to the Author):

Currently, there is a kind of 3rd or 4th renaissance in synthesizing, processing and applying BBL and related heteroaromatic ladder polymers. These polymers are inherently insoluble in organic solvents (or hardly soluble) and must be processed via special procedures, e.g. by casting films from high boiling, acidic solvents as the classical variant, as known for long time (Arnold, van Deusen, *Macromolecules* 1969). Some years later (1971) the same authors published a film forming process by drying (nano)particular BBL suspensions under film formation (Arnold, van Deusen, *J. Appl. Polym. Sci.* 1971, 15, 2035). These pioneering papers are not cited. This has clearly some flavour of ignoring the roots of the BBL topic. Later on, other groups did continuously work on BBL, including Sam Jenekhe (Seattle) and his group (the corresponding author published BBL stuff together with Sam Jenekhe), KS Narayan (Bangalore) and his group, and many others. These groups contributed many attractive results in the 1990s and 2000s, including device results (OFETs). The corresponding author of this manuscript is one driver in a new wave of BBL renaissance. He contributed some significant improvements in processing and handling of these very attractive heteroaromatic organic materials. Now, the authors describe a new, promising approach for obtaining blend films by blending and processing BBL (and other organo-insoluble conjugated polymers) with a conjugated polyelectrolyte (CPE), here an anionic polythiophene-based CPE with alkylencarboxylate side chains. Following this approach, ultrasonification of the polymer mixture leads to stable (micro)emulsions that can be processed into thin films. These films preserve the n-type semiconducting character of BBL and can be used as charge extracting interlayer of OSCs and active layer of OECTs. Varying the pH of the blends can initiate a kind of H-bond-formation-caused cross-linking and result in improved device stability.

I have the following remarks:

(1) Background: (a) As already said, BBL is introduced without any "historical" references. I am not sure if this is the best way. We all last on the shoulders of already done investigations. Today, the characterization and processing tools are, of course, much improved, but we can learn much by studying the "old" papers. (b) Also related solubilization approaches should be discussed in the introduction. The solubilization of ionic, water-insoluble conjugated polymers (CPs) by surfactants was extensively studied and described (e.g. by Matti Knaapila and co-workers as well as Andy Monkman and co-workers, e.g. in *J. Phys. Chem. B* 2006, 110, 10248-10257). The manuscript states that common surfactants do not help in emulsifying BBL. However, the selection of surfactants in the SI is a bit arbitrary. Moreover, PSS seems to have an effect on the BBL dispersions, simply by looking at the photo. (c) Also an embedding of neutral CPs into aqueous microemulsions was reported (very recently by Greek authors, see *J. Mater. Chem. B*, 2022, 10, 2680-2690), done in a ternary system water/oil/surfactant, and resulting in (nano)structures with hydrodynamic radii in the 10-70 nm range. (d) From a colloid science point of view, also other, related phenomena as coacervate formation may be discussed in the introduction, as described for blends of oppositely charged CPEs (Ayzner and co-workers, Sagelman and co-workers). Looking at Figure 1b one could assume the existence of some similarities to a coacervate formation. These colloid science-related points should be discussed in more detail. (e) Another literature-known

procedure for preparing conjugated polymer nanoparticles is the nanoprecipitation of CPs together with (ionic) block copolymers carrying carboxy groups (e.g. resulting from maleic anhydride as co-monomer), followed by (ultra)sonification (e.g. as described in *Angew. Chem. Int. Ed.* 2010, 49, 9436–9440). Also this process shows some similarities to the here described procedure. (f) OECT fabrication was described for blends of p- and n-type semiconducting ladder polymers (PBBTL/BBL), by authors from Singapore ("All-Polymer Bulk-Heterojunction Organic Electrochemical Transistors with Balanced Ionic and Electronic Transport", *Adv. Mater.* 2022, 34, 2206118). Their device results should be compared with the results of this investigation.

(2) The full role of the polythiophene-based conjugated polyelectrolyte is not really clear, especially its role for the electronic properties. It is simply a self-doped, ionic and polymeric surfactant that interacts with BBL? The here postulated ground state-energy transfer (as a central point, see the title of the manuscript) is, from my point of view) not fully supported by the experimental results. Fig. 2c/d is an only weak proof for a simultaneous doping of BBL and PCAT-K. The mathematical peak separation procedure and the associated errors remain largely unclear and unexplored. This is a central question since the peaks are very nearby located, with only minor shifts in the peak maxima and peak shapes of the XPS spectra. Cited from the manuscript (page 6) "From the peak areas, we estimated that 28.6% of BBL and 28.4% of PCAT-K are doped.": I am sceptical if such a precise estimation of the doping level(s) is possible based on these XPS results. The self-doping tendency of anionic polythiophene-based CPEs is well known (also in the absence of oxygen, see many results from the Gui Bazan group in US - UCSB) and is accompanied by redox processes of the polythiophene backbone, without any second component. Figure 2a: The low energy absorption shoulder may stand for CPE-based or BBL-based polarons. But: This must be investigated in more detail, e.g. explained and discussed with the help of optical polaron spectra of separately doped BBL and CPE homopolymers (p- or n-doped homopolymers). Based on the herein presented results, the origin of the low energy shoulder (most probably a CT band) in the optical spectra remains somewhat unclear. The emulsification process does not have an intimate link to the electronic properties.

(3) I also miss a detailed discussion of the molecular weights of CPEs and CPE precursors, and, if possible, also of BBL, and of the MW dependence of the emulsification process. By the way, it is well known, that ultrasonification leads to a significant amount of chain scissions. Also this point should be considered.

(4) Devices: What are the advantages of using charge extraction layers based on BBL blends in OSCs? Are there significant advantages in performance and stability (or price)? OECT: Can other BBL formulations cause similar device properties? Please compare with the results in *Adv. Mater.* 2022, 34, 2206118.

Reviewer #2 (Remarks to the Author):

This manuscript demonstrates that ground state electron transfer (GSET) between donor and acceptor polymers allows the processing of water-insoluble polymers from water. This approach enables macromolecular charge-transfer salts with 10,000× higher electrical conductivities than pristine polymers, low work function, and excellent thermal/solvent stability. This study will boost a new avenue

to develop water-based conductive inks for a wide range of potential applications in organic electronics. However, there are still several questions need to be addressed in the following aspects. Thus, we recommend this article can be accepted after minor revisions below:

1. In Supplementary Figure 2, the authors mentioned that “This results in smoother films compared to pure BBL processed from water, as visible by atomic force microscopy analysis.” To clarify the phase separation of the BBL: PCAT-K films, the authors should show the phase images in AFM.
2. In Supplementary Figure 3, the authors referred to figure (d) as out-of-plane and (e) as in-plane. The caption of the figure 3 is wrong.
3. Supplementary Figure 8a showed the UPS spectra of BBL, BBL/PCAT-K, and BBL/P3CPT-K. The authors should replace the Figure 3b with Supplementary Figure 8a.
4. The authors mentioned that GSET between donor and acceptor polymers allows the processing of water-insoluble polymers from water. Several papers reported the organic solar cell based on water soluble conjugated polymer. What is the novelty of this manuscript compared to the other papers? The novelty in the present manuscript should be clarified by the authors.

Reviewer #3 (Remarks to the Author):

In the manuscript, the authors demonstrated that ground-state electron transfer (GSET) between donor and acceptor polymers allows the processing of water-insoluble polymers from water. This approach enables macromolecular charge-transfer salts with 10,000× higher electrical conductivities than pristine polymers, low work function, and excellent thermal/solvent stability. They used these waterborne formulations to fabricate electron transport layers in non-fullerene OSCs, achieving power conversion efficiencies and operational stability superior to those using ZnO as the traditional electron transport layer. They also demonstrated high-performance OECTs, ultra-low power complementary OECT-based inverters, and OECNs with biorealistic spiking frequencies. In general, the manuscript is well written and extensive research has been carried out. Therefore, I suggest publication of the manuscript after some revisions.

1. Page 9. “Interestingly, we observed that polymers with shorter alkyl chains (i.e., p(g7NC2N) and p(g7NC4N)) dissolve well in the PCAT-K aqueous solution, while longer alkyl chains (i.e., p(g7NC6N) and p(g7NC8N)) cause the polymers to crash out of the PCAT-K aqueous solution (Supplementary Fig. 19).” The reason for this phenomenon should be provided.

2. The synthesis part in supplementary information: there are many typo/grammar errors. At the same time, full characterization of new materials should be provided.
3. Fig. 5b, Operational stability of OSCs with BBL:PCAT-K and ZnO as ETL under continuous light irradiation at AM 1.5G in an N₂-filled glovebox. Why only 25 h for the ZnO-based device? For comparison, the same test would be better.
4. For the insoluble BBL, how to know the molecular weight?

Response to Reviewers

Dear referees, we found your reviews to be very thoughtful and the comments extremely helpful in enhancing the quality, and thus the impact, of our manuscript. Below, please find our point-by-point response in red lettering to your concerns and a description of how and where revisions to the manuscript have been made.

Reviewer #1 (Remarks to the Author):

Currently, there is a kind of 3rd or 4th renaissance in synthesizing, processing and applying BBL and related heteroaromatic ladder polymers. These polymers are inherently insoluble in organic solvents (or hardly soluble) and must be processed via special procedures, e.g. by casting films from high boiling, acidic solvents as the classical variant, as known for long time (Arnold, van Deusen, Macromolecules 1969). Some years later (1971) the same authors published a film forming process by drying (nano)particular BBL suspensions under film formation (Arnold, van Deusen, J. Appl. Polym. Sci. 1971, 15, 2035). These pioneering papers are not cited. This has clearly some flavour of ignoring the roots of the BBL topic. Later on, other groups did continuously work on BBL, including Sam Jenekhe (Seattle) and his group (the corresponding author published BBL stuff together with Sam Jenekhe), KS Narayan (Bangalore) and his group, and many others. These groups contributed many attractive results in the 1990s and 2000s, including device results (OFETs). The corresponding author of this manuscript is one driver in a new wave of BBL renaissance. He contributed some significant improvements in processing and handling of these very attractive heteroaromatic organic materials. Now, the authors describe a new, promising approach for obtaining blend films by blending and processing BBL (and other organo-insoluble conjugated polymers) with a conjugated polyelectrolyte (CPE), here an anionic polythiophene-based CPE with alkylencarboxylate side chains. Following this approach, ultrasonification of the polymer mixture leads to stable (micro)emulsions that can be processed into thin films. These films preserve the n-type semiconducting character of BBL and can be used as charge extracting interlayer of OSCs and active layer of OECTs. Varying the pH of the blends can initiate a kind of H-bond-formation-caused cross-linking and result in improved device stability.

We appreciate the reviewer's thoughtful feedback on our work, which has provided us with valuable insights for enhancing the quality of our manuscript. Before delving into the technical points raised by the reviewer, we want to emphasize that our primary focus was not to conduct an exhaustive survey of the literature pertaining to BBL. Instead, our objective is to introduce a novel approach aimed at simultaneously improving the solubility and conductivity of water-insoluble conjugated polymers. Our results illustrate that ground-state electron transfer in all-polymer donor:acceptor blends enables the dispersion of water-insoluble conjugated polymers in water. We selected BBL as an illustrative example of a water-insoluble conjugated polymer with a high electron affinity. Importantly, we have demonstrated that this approach is general and can be extended to other high-performance water-insoluble conjugated polymers, as exemplified by several side chain-free ladder-type polymers, as well as lactam-based and benzodifurandione-based polymers featuring side chains with varying degrees of hydrophilicity. Therefore, our intention was not to disregard or conceal prior literature on BBL but rather to provide the reader with the information necessary to contextualize our work within the broader literature landscape.

In the following, we address the reviewer's remarks:

(1) Background: (a) As already said, BBL is introduced without any "historical" references. I am not sure if this is the best way. We all last on the shoulders of already done investigations. Today, the characterization and processing tools are, of course, much improved, but we can learn much by studying the "old" papers.

We have included references to the pioneering historical papers that cover the synthesis, processing, and applications of BBL. Specifically, the following references have been added as Refs. 44-47: *Macromolecules*, 1969, 2, 497-502; *J. Appl. Polym. Sci.*, 1971, 15, 2035-2047; *J. Am. Chem. Soc.*, 2003, 125, 13656-13657; *J. Appl. Phys.*, 1995, 77, 3938-3941).

(b) Also related solubilization approaches should be discussed in the introduction. The solubilization of ionic, water-insoluble conjugated polymers (CPs) by surfactants was extensively studied and described (e.g. by Matti Knaapila and co-workers as well as Andy Monkman and co-workers, e.g. in *J. Phys. Chem. B* 2006, 110, 10248-10257). The manuscript states that common surfactants do not help in emulsifying BBL. However, the selection of surfactants in the SI is a bit arbitrary. Moreover, PSS seems to have an effect on the BBL dispersions, simply by looking at the photo.

The solubilization of ionic, water-insoluble conjugated polymers by means of surfactants and polyelectrolytes had already been described in the introduction of the original manuscript (p.2-3). Nevertheless, we thank the reviewer for pinpointing the very pertinent work by Knaapila and Monkman (*J. Phys. Chem. B* 2006, 110, 10248-10257) that we had overlooked in our survey of the field. This work is now referenced in the revised manuscript as Ref. 24.

Regarding the selection of surfactants reported in the Supplementary Information, these were chosen as an example of anionic, cationic, and non-ionic surfactants. It should be noted that these surfactants have already been used to disperse other insoluble carbon-based materials (e.g., conducting polymer, graphene) in water, as reported in Refs. 1-6 of Supplementary Information. Importantly, these surfactants have similar functional groups to PCAT-K: PEG and TWEEN80 have alkoxy groups, while PAA, PSS, and PFI have anionic (carboxylic and sulfonic) groups. PEI was chosen as an example of a polycationic surfactant that is capable of dispersing BBL in alcohol (*Nat. Commun.* 2021, 12, 2354). As shown in Suppl. Figure 3, none of these surfactants can disperse BBL in water. This observation is corroborated by the measured BBL:surfactant particle size, which consistently ranges from 400 to 2000 nm (Suppl. Figure 4). This is comparable to the particle size of pristine BBL dispersion without surfactants (400-800 nm) and notably larger than the BBL:PCAT-K nanoparticles (~30 nm). This trend is visually supported by the images of spin-cast BBL:surfactant films on glass, as shown in Suppl. Figure 5, where inhomogeneous films with large aggregates are evident. For comparison, BBL:PCAT-K shows homogeneous films.

(c) Also an embedding of neutral CPs into aqueous microemulsions was reported (very recently by Greek authors, see *J. Mater. Chem. B*, 2022, 10, 2680-2690), done in a ternary system water/oil/surfactant, and resulting in (nano)structures with hydrodynamic radii in the 10-70 nm range.

We thank the reviewer for pinpointing the interesting work by Mitsou et al. in *J. Mater. Chem. B*, 2022, 10, 2680-2690. While the work of Mitsou and co-workers shows an interesting strategy to form aqueous microemulsions of conjugated polymers by using ternary water/oil/surfactant systems, it has several differences with ours:

1) It still relies on flammable and hazardous organic solvents (e.g., 3.8 wt% limonene) to form the aqueous microemulsions, while our approach only requires water.

2) The surfactant accounts for most of the mass, and the dispersion has a polymer load in the range of $\mu\text{g/mL}$ ($< 0.001\%$ of the total mass in the microemulsion). These very diluted aqueous microemulsions pose a challenge for the solution processing step. Our approach yields solutions with polymer loading in the g/L range.

3) The surfactant (Span 80, Labrasol[®]) is non-conductive and remains in the solid-state film, leading to poor electrical properties. In our case, PCAT-K not only helps BBL disperse in water but also yields a 10000-fold increase in the electrical conductivity of the blend due to the spontaneous ground-state electron transfer.

Nevertheless, we found the paper by Mitsou et al. pertinent to our study and reference it in the introduction of the revised manuscript as Ref. 26.

(d) From a colloid science point of view, also other, related phenomena as coacervate formation may be discussed in the introduction, as described for blends of oppositely charged CPEs (Ayzner and co-workers, Sagelman and co-workers). Looking at Figure 1b one could assume the existence of some similarities to a coacervate formation. These colloid science-related points should be discussed in more detail.

This is an excellent suggestion. We discussed the phenomenon of coacervate formation in the introduction (p.3) and referenced the works of Ayzner et al. (*J. Phys. Chem. Lett.*, 2022, 13, 44, 10275–10281) and Segalman et al. (*ACS Macro Lett.*, 2021, 10, 8, 1008–1014) as refs. 30 and 31.

(e) Another literature-known procedure for preparing conjugated polymer nanoparticles is the nanoprecipitation of CPs together with (ionic) block copolymers carrying carboxy groups (e.g. resulting from maleic anhydride as co-monomer), followed by (ultra)sonification (e.g. as described in Angew. Chem. Int. Ed. 2010, 49, 9436–9440). Also this process shows some similarities to the here described procedure.

As correctly pointed out by the reviewer, the preparation of conjugated polymer nanoparticles by precipitation of conjugated polymer with (ionic) block copolymers carrying carboxyl groups is known and shares some similarities to the process described in our manuscript. However, it should be noted that there are several compelling differences that set our work aside from those reported in the literature:

1) Most conjugated polymer nanoparticle dispersions reported in the literature are in the range of $\mu\text{g/ml}$ (e.g., 50 $\mu\text{g/ml}$ in the work cited by the reviewer *Angew. Chem. Int. Ed.*, 2010, 49, 9436-9440). These very diluted dispersions are typically used in spectroscopic investigations (e.g., fluorescent probes in the work above, now referenced as Ref. 27) and are hardly employed in the fabrication of organic electronic devices. The latter requires the use of dispersions with concentrations in the range of mg/mL. In our work, the maximum concentration of BBL:PCAT-K solution is about 8 mg/ml. This allowed us to use BBL:PCAT-K ink to successfully fabricate organic solar cells and OECTs as well as other organic electrochemical devices.

2) As already noted in our answer to Q1-c above, most surfactants or (ionic) block copolymers used to form conjugated polymer nanoparticles are non-conductive and will remain in the solid-state films, adversely impacting the materials' electrical properties and relative devices' metrics.

In our case, PCAT-K not only assists BBL in dispersing in water but also yields a 10000-fold increase in electrical conductivity due to the spontaneous ground-state electron transfer process.

(f) OECT fabrication was described for blends of p- and n-type semiconducting ladder polymers (PBBTL/BBL), by authors from Singapore ("All-Polymer Bulk-Heterojunction Organic Electrochemical Transistors with Balanced Ionic and Electronic Transport", Adv. Mater. 2022, 34, 2206118). Their device results should be compared with the results of this investigation.

The work cited by the reviewer (Adv. Mater. 2022, 34, 2206118) describes the development of OECTs in which p-type (PBBTL) and n-type (BBL) conjugated ladder polymers are processed together from methanesulfonic acid to form ambipolar blends. The work cited by the reviewer is thus fundamentally different from our study: 1) The energetics of PBBTL and BBL do not favor ground-state charge transfer between the two polymers, such that the PBBTL:BBL blends are non-conductive. 2) PBBTL and BBL are both processed from strong acids. In stark contrast, our study shows that ground-state electron transfer between the water-insoluble BBL and the water-soluble PCAT-K enables the dispersion of BBL in water. The crucial and unprecedented advancement with our BBL:PCAT-K formulation compared to the other BBL:polymer blends is that our aqueous ink eliminates the need for a highly acidic solvent, thus making it suitable for large-scale applications. Nonetheless, in order to address the reviewer's comment, we have summarized the performance with different BBL formulations in Suppl. Table 4.

(2) The full role of the polythiophene-based conjugated polyelectrolyte is not really clear, especially its role for the electronic properties. It is simply a self-doped, ionic and polymeric surfactant that interacts with BBL? The here postulated ground state-energy transfer (as a central point, see the title of the manuscript) is, from my point of view) not fully supported by the experimental results. Fig. 2c/d is an only weak proof for a simultaneous doping of BBL and PCAT-K. The mathematical peak separation procedure and the associated errors remain largely unclear and unexplored. This is a central question since the peaks are very nearby located, with only minor shifts in the peak maxima and peak shapes of the XPS spectra. Cited from the manuscript (page 6) "From the peak areas, we estimated that 28.6% of BBL and 28.4% of PCAT-K are doped.": I am sceptical if such a precise estimation of the doping level(s) is possible based on these XPS results. The self-doping tendency of anionic polythiophene-based CPEs is well known (also in the absence of oxygen, see many results from the Gui Bazan group in US - UCSB) and is accompanied by redox processes of the polythiophene backbone, without any second component. Figure 2a: The low energy absorption shoulder may stand for CPE-based or BBL-based polarons. But: This must be investigated in more detail, e.g. explained and discussed with the help of optical polaron spectra of separately doped BBL and CPE homopolymers (p- or n-doped homopolymers). Based on the herein presented results, the origin of the low energy shoulder (most probably a CT band) in the optical spectra remains somewhat unclear. The emulsification process does not have a intimate link to the electronic properties.

We thank the reviewer for this thought-provoking comment. We have shown that not every polythiophene-based conjugated polyelectrolyte can disperse BBL in water. We selected PCAT-K and P3CPT-K as two examples of polythiophenes bearing carboxylic groups but having different energy levels (as shown in Suppl. Figure 13). It should be noted that both PCAT-K and P3CPT-K are in the neutral state (not self-doped) as confirmed by the absence of characteristic spectroscopic features of positive polarons in the UV-vis-NIR spectra of the pristine films (see Figure 2a and Suppl. Figure 10b for PCAT-K and Suppl. Figure 15d for P3CPT-K), as well as their low electrical conductivity (Suppl. Figure 18). Of the two polymers, only PCAT-K proved to efficiently disperse BBL in water, a phenomenon we attributed to ground-state charge transfer.

We have demonstrated the occurrence of ground-state electron transfer between PCAT-K and BBL using a variety of different techniques. UPS measurements reveal a significant energy level shift when PCAT-K is deposited on top of BBL (Suppl. Figure 13), indicating a substantial electron transfer from PCAT-K to BBL. Interestingly, P3CPT-K on BBL shows a negligible energy level shift, supporting the absence of significant ground-state electron transfer. Both UV-Vis-NIR and EPR measurements confirm the presence of polarons in the BBL:PCAT-K blend films (Figure 2a-b). In addition, FT-IR (Suppl. Figure 12) and UV-Vis-NIR spectra (Suppl. Figure 10) of chemically doped BBL and PCAT-K show similar spectroscopic features to those observed for BBL:PCAT-K blends. All these measurements provide additional evidence for the presence of both doped BBL and PCAT-K chains as observed by XPS.

For the analysis of the XPS data, we conducted a thorough examination of peak deconvolution. The fitting of BBL N1s was based on the analysis done by Chen et al. (*Macromolecules*, 2022, 55, 16, 7294–7302) and Nalwa et al. (*Polymer*, 1991, 32, 802-807). Similarly, the fitting of PCAT-K was based on the work of Greczynski et al. (*J. Electron Spectrosc. Relat. Phenom.*, 2001, 121, 1) and Xing et al. (*Synthetic Metals*, 1997, 89, 161-165). For the pristine BBL and PCAT-K films, the XPS data can be accurately fitted using only one peak (solid line in Figure 2c-d), indicating the presence of only one chemical state (neutral) for N and S. However, after blending BBL and PCAT-K, a noticeable peak shift is observed, and the measured results cannot be fitted with a single peak, as was the case with the pristine film. This suggests the existence of mixed chemical states (neutral and charged) for the N or S atoms in the blend film (dash line in Figure 2c-d). Considering that N is exclusively from BBL and S is exclusively from PCAT-K, we can infer that there is partial doping of both BBL and PCAT-K. We determined the doping level from the ratio of the peak areas corresponding to different chemical states in the doped polymers. Note that this is a common way to estimate the doping level of doped polymers (e.g., see *Chem. Mater.* 2016, 28, 10, 3462–3468 or *Nat. Mater.*, 2011, 10, 429–433). While the precision of the peak fitting may not allow for an exact doping level estimation down to a tenth of a percent, in our case, we were still able to qualitatively estimate a very similar doping level (ca. 28%) for both BBL and PCAT-K within the BBL:PCAT-K blends from the XPS data with a high degree of reliability. This further supports the notion of doping through charge transfer from PCAT-K to BBL.

As mentioned earlier, we conducted UV-Vis-NIR spectroscopy on chemically doped BBL and PCAT-K (Suppl. Figure 10). BBL was doped using the reductant PEI, while PCAT-K was doped using the oxidant FeCl₃. Upon doping, we observed a quenching of the ground state absorption and the emergence of distinct n/p polaron absorption features in both polymers. BBL exhibits characteristic polaronic peaks at low wavelengths (700-1000 nm), while PCAT-K shows several broad features at both lower (800-1100 nm) and higher wavelengths (>1200 nm). We noted that BBL's polaron absorption shows a distinct peak at 710 nm along with a vibrational progression, which is also evident in the BBL:PCAT-K spectra. However, the peak ratio differs significantly in the BBL:PCAT-K spectra. The absorption peak at higher wavelengths (*i.e.*, 880 nm) is more pronounced compared to PEI-doped BBL, reminiscent of the polaron spectrum of doped PCAT-K. Hence, we concluded that the differential spectra of BBL:PCAT-K blend are dominated by the BBL n-polaron but also include contributions from the PCAT-K p-polaron. Furthermore, we conducted FTIR measurements of chemically doped BBL and PCAT-K and observed a similar peak shift as seen in the BBL:PCAT-K blend. This strongly indicates the coexistence of both n-doped BBL and p-doped PCAT-K within the blend.

The new UV-Vis-NIR and FT-IR spectra of chemically doped BBL and PCAT-K are added as Suppl. Figure 10 and Suppl. Figure 12 in the revised manuscript.

(3) I also miss a detailed discussion of the molecular weights of CPEs and CPE precursors, and, if possible, also of BBL, and of the MW dependence of the emulsification process. By the way, it is well known, that ultrasonification leads to a significant amount of chain scissions. Also this point should be considered.

Excellent comment. We studied the effect of the molecular weight of PCAT-K and BBL on the dissolution process of BBL in water. We observed that high molecular weight PCAT-K can help disperse more BBL, while high molecular weight BBL requires more PCAT-K to disperse effectively in water. The relationship between BBL:PCAT-K ratio and the molecular weight of the polymers is reported in Suppl. Figure 42. This indicates that high molecular weight PCAT-K, with its longer chain and increased flexibility, has a higher degree of freedom to interact with BBL compared to its lower molecular weight counterpart, thus aiding in the dissolution of more BBL in water. It is worth noting that the molecular weight of P3CPT-K is about 55 kDa, which is significantly larger than both PCAT-K and BBL. Nevertheless, P3CPT-K is unable to disperse BBL in water. This suggests that ground state electron transfer exerts a more dominant influence than molecular weight itself in the case of conjugated polyelectrolytes.

Addressing the concern about ultrasonication potentially causing polymer chain scissions, we conducted measurements of the molecular weight of BBL and PCAT-K under continuous sonication (180 W, 30 min) and observed no changes in the molecular weight of both polymers (see Suppl. Figure 41).

(4) Devices: What are the advantages of using charge extraction layers based on BBL blends in OSCs? Are there significant advantages in performance and stability (or price)? OEET: Can other BBL formulations cause similar device properties? Please compare with the results in *Adv. Mater.* 2022, 34, 2206118.

Figures 4a-b demonstrate that employing a BBL:PCAT-K blend as the electron transport layer (ETL) in non-fullerene organic solar cells leads to devices with significantly enhanced stability compared to those using a ZnO-based ETL. While ZnO is a commonly used ETL in organic solar cells, it is prone to instability when exposed to UV light (e.g., see *Nat. Commun.*, 2021, 12, 5419). Therefore, we attributed the enhanced operational stability of the non-fullerene solar cells incorporating the BBL:PCAT-K-based ETL to the superior UV stability exhibited by the BBL:PCAT-K blend.

Regarding the OEET application, Figures 4c-f show that we were able to develop n-type OEETs using BBL entirely processed from water. These devices exhibit performance comparable to conventional BBL-based OEETs processed using concentrated methanesulfonic acid (refer to Supplementary Figure 31 and Supplementary Table 4 for a survey of the field). The crucial advancement with our BBL:PCAT-K blend formulation compared to the traditional BBL is that our aqueous ink eliminates the need for a highly acidic solvent, thus making it suitable for large-scale applications. It should be noted that the work cited by the reviewer (*Adv. Mater.* 2022, 34, 2206118) still refers to the processing of BBL from concentrated methanesulfonic acid (as noted in our answer to Q1-f above).

Reviewer #2 (Remarks to the Author):

This manuscript demonstrates that ground state electron transfer (GSET) between donor and acceptor polymers allows the processing of water-insoluble polymers from water. This approach enables macromolecular charge-transfer salts with $10,000\times$ higher electrical conductivities than pristine polymers, low work function, and excellent thermal/solvent stability. This study will boost a new avenue to develop water-based conductive inks for a wide range of potential applications in organic electronics. However, there are still several questions need to be addressed in the following aspects. Thus, we recommend this article can be accepted after minor revisions below:

We thank the reviewer for the positive commentary on our manuscript and for considering our work impactful and innovative. In the following, we address their remarks:

1. In Supplementary Figure 2, the authors mentioned that “This results in smoother films compared to pure BBL processed from water, as visible by atomic force microscopy analysis.” To clarify the phase separation of the BBL:PCAT-K films, the authors should show the phase images in AFM.

We have incorporated updated AFM images of BBL, PCAT-K, and BBL:PCAT-K films with varying ratios (found in Suppl. Figure 6). The BBL:PCAT-K blend films, across different ratios, exhibit significantly smoother surfaces compared to the pristine BBL film. The phase images of the BBL:PCAT-K blend films do not reveal clear indications of phase separation.

2. In Supplementary Figure 3, the authors referred to figure (d) as out-of-plane and (e) as in-plane. The caption of the figure 3 is wrong.

We thank the reviewers for pointing out this mistake. We have corrected Suppl. Figure 3 (now Suppl. Figure 7) in the revised manuscript.

3. Supplementary Figure 8a showed the UPS spectra of BBL, BBL/PCAT-K, and BBL/P3CPT-K. The authors should replace the Figure 3b with Supplementary Figure 8a.

Again, excellent comment. The UPS spectra of BBL, BBL/PCAT-K and BBL/P3CPT-K, shown in Suppl. Figure 8a (now Suppl. Figure 13 of the revised manuscript), were used to show the energy level shifts and to confirm the occurrence of the charge transfer process between BBL and PCAT-K. To address the reviewer’s comment, we have revised Figure 3b, including the UPS spectra of BBL and BBL/PCAT-K, clearly showing the work function shift. We have decided to leave the UPS spectra of BBL/P3CPT-K in Suppl. Figure 13 (formerly Suppl. Fig. 8a) together with the UPS spectra of BBL and BBL/PCAT-K, as these data are discussed earlier on in the main text and only supports the evidence that P3CPT-K does not yield significant ground-state electron transfer when in contact with BBL due to its high ionization energy.

4. The authors mentioned that GSET between donor and acceptor polymers allows the processing of water-insoluble polymers from water. Several papers reported the organic solar cell based on water soluble conjugated polymer. What is the novelty of this manuscript compared to the other papers? The novelty in the present manuscript should be clarified by the authors.

We acknowledge the reviewer’s feedback regarding clarity. Most water-soluble conjugated polymers are based on polythiophene or poly(3,4-ethylenedioxythiophene) and are typically

used as hole-transport layers due to their high work function. In the realm of waterborne electron-transport layers for organic solar cells, metal-oxide nanoparticles (such as ZnO and SnO₂), organic small molecules, and polyelectrolytes are the predominant choice. There are only a few instances in the literature that explore water-soluble conjugated polymers with low work function, primarily based on polyfluorene and naphthalenediimide. In comparison to these materials, BBL:PCAT-K presents several distinctive features:

1) BBL:PCAT-K has a relatively high conductivity of about 0.35 S/cm. The materials reported in the literature are predominantly undoped or self-doped (slightly doped), with conductivities much lower, typically below 10⁻⁵ S/cm.

2) Previous materials in the literature often have limited solubility in water, necessitating the use of alcohol as a co-solvent. In the case of BBL:PCAT-K ink, only water is used as the solvent.

We have compiled a summary of the water/alcohol-soluble conjugated polymer used as electron transport layers for comparison in the revised manuscript (Suppl. Table 3).

Reviewer #3 (Remarks to the Author):

In the manuscript, the authors demonstrated that ground-state electron transfer (GSET) between donor and acceptor polymers allows the processing of water-insoluble polymers from water. This approach enables macromolecular charge-transfer salts with 10,000× higher electrical conductivities than pristine polymers, low work function, and excellent thermal/solvent stability. They used these waterborne formulations to fabricate electron transport layers in non-fullerene OSCs, achieving power conversion efficiencies and operational stability superior to those using ZnO as the traditional electron transport layer. They also demonstrated high-performance OECTs, ultra-low power complementary OECT-based inverters, and OECNs with biorealistic spiking frequencies. In general, the manuscript is well written and extensive research has been carried out. Therefore, I suggest publication of the manuscript after some revisions.

We thank the reviewer for the very positive commentary of our manuscript and for considering this work innovative. In the following, we address their remarks:

1. Page 9. “Interestingly, we observed that polymers with shorter alkyl chains (i.e., p(g7NC2N) and p(g7NC4N)) dissolve well in the PCAT-K aqueous solution, while longer alkyl chains (i.e., p(g7NC6N) and p(g7NC8N)) cause the polymers to crash out of the PCAT-K aqueous solution (Supplementary Fig. 19).” The reason for this phenomenon should be provided.

This is an excellent comment. The series of lactam polymers p(g₇NC_nN) we tested in our study have identical backbone and oligo ethylene glycol side chains but gradually increasing alkyl chain lengths. This results in an overall increasing polymer chain hydrophobicity, as confirmed by contact angle measurements. We speculated that the increased hydrophobicity of the polymer chains affects the water stability of the p(g₇NC_nN):PCAT-K complex, causing the polymers to

aggregate and crash out of the aqueous solution. In revision, we discuss this phenomenon in more detail and provide the contact angle data in a new Suppl. Figure 25.

2. *The synthesis part in supplementary information: there are many typo/grammar errors. At the same time, full characterization of new materials should be provided.*

We have carefully reviewed and addressed the concerns raised in the synthesis part of the SI. We have also provided a comprehensive characterization of PCAT-K, including ¹H-NMR, elemental analysis, and GPC measurements in the Supplementary Information.

3. *Fig. 5b, Operational stability of OSCs with BBL:PCAT-K and ZnO as ETL under continuous light irradiation at AM 1.5G in an N₂-filled glovebox. Why only 25 h for the ZnO-based device? For comparison, the same test would be better.*

Again, excellent comments. Given that ZnO-based devices experience a rapid decline to below 80% of their initial performance (referred to as T₈₀) and that T₈₀ is a key factor in estimating the lifetime of a solar cell [e.g., see *Nature Energy* 5, 35-49 (2020)], we had initially opted to only present data up to T₈₀. In revision, we present the operational stability of ZnO-based and BBL:PCAT-K-based devices with the same illumination time for better comparison.

4. *For the insoluble BBL, how to know the molecular weight?*

The molecular weight of BBL is estimated by measuring the intrinsic viscosity of BBL solution in MSA using the approach reported by Berry et al. [*Addition and Condensation Polymerization Processes, Vol. 91 (Ed: N. A. J. Platzer), American Chemical Society, Washington, DC, USA 1969, p. 734*]. Here, it is assumed that the viscosity (η) and molecular weight (M) follow the Mark-Houwink-Sakurada equation $\eta = KM^\alpha$, where the K and α are $5.11 \times 10^{-6} \text{ g dL}^{-1}$ and $\alpha = 1.34$, respectively. We have revised the experimental section to address this comment.

REVIEWERS' COMMENTS

Reviewer #1 (Remarks to the Author):

After reading the point-by-point reply letter and the carefully revised manuscript file I can now recommend acceptance of the current version.

One final remark: In the figures, the lines of the graphs are sometimes difficult to recognise, the line width is chosen too small and some of the colours are poorly reproduced (e.g. light blue).

Reviewer #2 (Remarks to the Author):

The authors have revised manuscript appropriately to the reviewers' comments and so I recommend the publication of this article without further revision.

Reviewer #3 (Remarks to the Author):

The authors have addressed my concerns and quality of the manuscript has been improved. Therefore, I suggest publication of the manuscript.

Response to Reviewers

Reviewer #1 (Remarks to the Author):

After reading the point-by-point reply letter and the carefully revised manuscript file I can now recommend acceptance of the current version.

One final remark: In the figures, the lines of the graphs are sometimes difficult to recognise, the line width is chosen too small and some of the colours are poorly reproduced (e.g. light blue).

Thank you for helping us improve the quality of our manuscript. As suggested by the reviewer, we have revised the figures to improve readability.

Reviewer #2 (Remarks to the Author):

The authors have revised manuscript appropriately to the reviewers' comments and so I recommend the publication of this article without further revision.

Thank you for helping us improve the quality of our manuscript.

Reviewer #3 (Remarks to the Author):

The authors have addressed my concerns and quality of the manuscript has been improved. Therefore, I suggest publication of the manuscript.

Thank you for helping us improve the quality of our manuscript.